

# Extracellular vesicles in endometrial-related diseases: role, potential and challenges

Zilu Wang[1], Feng Li[1] and Wenqiong Liu[2]

[1] Shandong University of Traditional Chinese Medicine, Jinan, China
[2] Shandong Provincial Hospital of Traditional Chinese Medicine, Jinan, China

## ABSTRACT

Endometrial dysfunction underlies many common gynecologic disorders, such as endometriosis, endometrial cancer, intrauterine adhesions, and endometritis, which affect many women around the world. Extracellular vesicles play an important role in the pathophysiologic process of endometrial-related diseases. Extracellular vesicles are released by cells, which usually act as a form of intercellular communication, affecting biological processes such as fibrosis, angiogenesis, cell proliferation, and inflammatory responses by transferring their own proteins, lipids, RNA transcripts, and DNA for messaging, and play a key role in physiological dynamic homeostasis and disease development. This review combines the studies of the last decade, using the sub-description method to introduce the application of different sources of extracellular vesicles in the diagnosis and treatment of related diseases, and discusses the challenges faced by extracellular vesicles in the diagnostic and therapeutic application of endometriosis-related diseases, with the aim of contributing to our understanding of the mechanism of action of extracellular vesicles and their therapeutic roles, so as to provide a reference for the development of endometriosis-related diseases, as well as their prognosis and treatment.

## INTRODUCTION

Extracellular vesicles (EVs) are released by cells, which usually act as a form of intercellular communication, and consist mainly of exosomes, microvesicles, and apoptotic bodies, where exosomes (30–150 nm) are derived from the endosomal pathway, and are intraluminal vesicles released from the plasma membrane by fusion of the multivesicular body (MVB) with the plasma membrane into the extracellular environment, apoptotic bodies (100–1,500 nm) are produced by plasma membrane release, and microvesicles are formed from plasma membrane lipid rafts that sprout and split outwards (*Villarroya-Beltri et al., 2014*; *Nguyen et al., 2016*; *Mathieu et al., 2019*). Most cells such as endometrial cells, placental cells, reticulocytes, tumor cells, hematopoietic cells, dendritic cells, intestinal epithelial cells, and neurons secrete various types of extracellular vesicles (*Ng et al., 2013*; *Sarker et al., 2014*; *Théry et al., 2002*; *Van Niel et al., 2001*). EVs play an important role in

Corresponding author
Wenqiong Liu, 2621504134@qq.com

a range of pathophysiologic processes. This is because extracellular vesicles can transfer specific substances, such as proteins, lipids, RNA transcripts and DNA, to target cells, which in turn can alter and influence the function of the target cells (*Xu et al., 2016*). EVs act as carriers that can deliver the contained molecules from one cell or tissue to another through protein interactions, endocytosis, or direct membrane fusion, and protect their contents from degradation or modification by extracellular enzymes, and then exert biological effects by binding to receptors on the surface of the recipient cell or delivering their contents by cytosolization and cytosol delivery (*Colombo, Raposo & Théry, 2014*). However, the mechanism of information transfer in EVs is still not fully defined. Extracellular vesicles from different cells and phenotypes contain different lipids, proteins, mRNA, miRNA, and DNA, and these specific intracellular contents can be transferred to their target cells, where they fulfill their functions (*Raposo & Stoorvogel, 2013*; *Rana & Zöller, 2011*; *Mulcahy, Pink & Carter, 2014*). Among the molecular substances carried by EVs, miRNA is a widely studied class of inclusion. miRNAs in EVs (EV-miRNAs) act as signaling molecules to mediate intercellular communication and play important roles in physiological and pathological processes.

The endometrial layer (endometrium, uterine endometrium) is a layer of mucous membrane that makes up the inner wall of the mammalian uterus and, unlike other mucous tissues, undergoes a dynamic, cyclical process of shedding, regeneration, and differentiation throughout reproduction that is coordinated by the hypothalamic-pituitary-ovarian axis. Endometrial dysfunction underlies many common gynecological disorders such as endometriosis, abnormal uterine bleeding, miscarriage, infertility, pre-eclampsia and endometrial cancer, which affect many women around the world (*Garcia-Alonso et al., 2021*; *Garrido-Gomez et al., 2017*; *Rabaglino & Conrad, 2019*; *Salker et al., 2011*; *Houshdaran et al., 2020*). Studies have shown that EVs can be released from human endometrial epithelial cells, chorionic trophoblasts, efferent trophoblasts and placental trophoblasts (*Ng et al., 2013*; *Luo et al., 2009*; *Donker et al., 2012*). They are also closely related to endometrial diseases and have an inhibitory effect on endometrial diseases and promote the growth and repair of the endometrium, and there are also some related extracellular vesicles that play a promotional role in the development of endometrial diseases (*Zhang et al., 2024*). The study of the relationship between extracellular vesicles and the endometrium is important for the treatment of endometrial disease, which will help to deepen the understanding of endometrial disease and improve the prognosis of endometrial disease (Fig. 1).

## SURVEY METHODOLOGY

Literature searches were conducted in the PubMed, Web of Science. In addition to articles published since 2015, earlier articles were also considered. The keywords used were as follows: extracellular vesicles, extracellular vesicles and endometriosis/infertility/intrauterine adhesion/endometrial carcinoma/endometritis. As the work progressed, we conducted a literature search using the keywords extracellular vesicles and treatment/diagnosis/challenges. However, non-research articles or reviews were excluded.

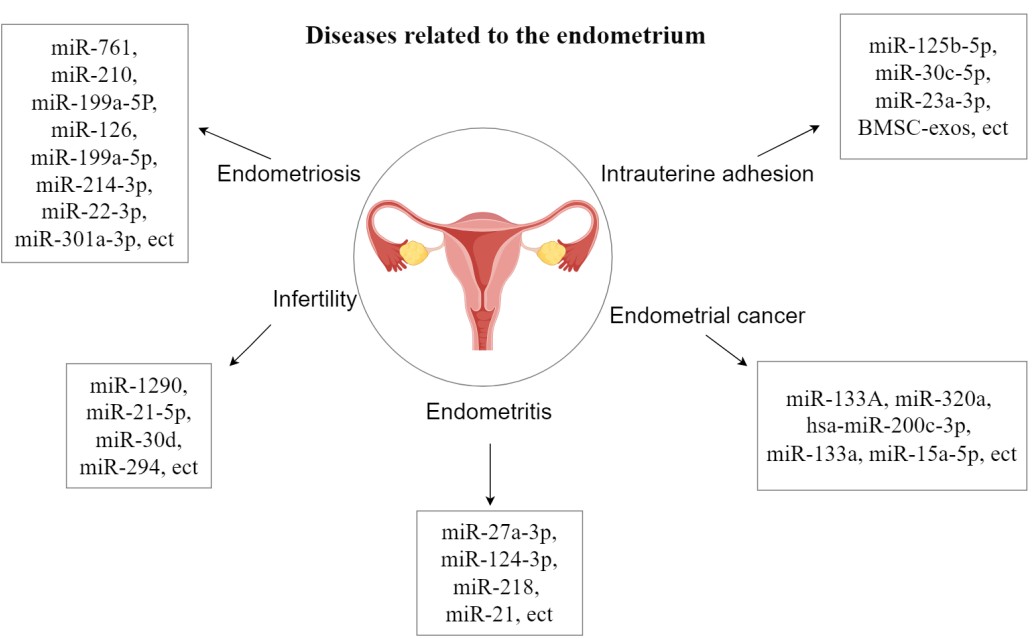

Figure 1  Diseases related to the endometrium.

After removing duplicate articles and the articles with little relevance, 108 articles were selected for this review.

## Endometriosis

Endometriosis is a benign and hormone-dependent disease caused by ectopic deposition of endometrial tissue in the abdominal cavity and outside the uterus. The main clinical manifestations are infertility, dysmenorrhea, irregular menstruation, and discomfort during sexual intercourse (*Asghari et al., 2018*). Studies have shown that women with endometriosis have significant differences in long chain non-coding RNAs, miRNAs, and proteins involved in histone modification, angiogenesis, and immunomodulation contained in endometrial cell exosomes compared to normal women, and it has also been experimentally demonstrated that exosomes and their vectors play important roles in angiogenesis, neuromodulation, immunomodulation, and endometrial stromal cell invasion (*Freger, Leonardi & Foster, 2021*; *Khalaj et al., 2019*; *Zhou et al., 2020*; *Wu et al., 2021*). Thus, it is suggested that exosomes play an important role in the pathophysiology of endometriosis.

### EVs inhibit fibrosis formation

Although the exact cause of endometrial fibrosis is unknown, a number of studies have shown that connective tissue growth factors are crucial for the production of fibers, and fibrosis is a significant cause of pain and infertility. The broad anti-fibrotic efficacy of connective tissue growth factor (CCN2) has been shown *in vitro* and *in vivo* in experimental fibrosis models, making it an appealing therapeutic target (*Brigstock, 2009*). MiRNAs are the primary RNA component of exosomes and are recognized as potent controllers

of gene expression (*Redis et al., 2012*). By attaching to complementary locations in the target mRNA's 3′ untranslated region (UTR), they control the expression of target genes, causing translational repression, mRNA degradation, and gene silencing (*Guo et al., 2010*). *Zhang et al. (2021b)* delved into the exosomes derived from endometrial stromal cells in endometriosis patients with the help of a dual validation strategy of *in vivo* experiments and *in vitro* models. They found that the expression level of miR-214-3p was significantly reduced in these exosomes, while the expression of CCN2 showed an up-regulation trend. This important finding not only reveals that miR-214-3p in exosomes can effectively curb the pathological process of endometrial fibrosis by precisely regulating the expression of CCN2, but also further highlights the great potential of miR-214-3p as a biomarker in the diagnosis of endometriosis (EMS) and its broad application prospects as a novel therapeutic target in the treatment strategy of endometriosis. broad application prospects. Studies by *Wu et al. (2018a)* found that upregulation of miR-214 in exosomes may inhibit fiber formation and its delivery by exosomes derived from ectopic endometrial stromal cells.

### EVs promote angiogenesis

Angiogenesis is a necessary process for the formation and establishment of endometriotic lesions. Exosomes derived from endometriotic stromal cells have been shown to effectively promote angiogenesis (*Harp et al., 2016*). *Sun et al. (2019a)* also found that ectopic endometrium with endometriosis secretes exosomes with a diameter of approximately 100 nm, which are internalized by human umbilical vein endothelial cells and DRG neurons and can enhance angiogenesis. After cultivating human umbilical vein endothelial cells (HUVECs) using exosomal homodimer transcription of the antisense RNA HOTAIR, *Zhang et al. (2022a)* observed improved angiogenesis. Additionally, homeobox transcript antisense RNA (HOTAIR) down-regulated miR-761 and up-regulated the expression of histone deacetylase 1 (HDAC1). On the other hand, exosomal HOTAIR's function in endometrial stromal cells and HUVECs was reversed by miR-761 overexpression or HDAC1 knockdown. The exocrine HOTAIR is packaged and transported from endometrial stromal cells to surrounding cells. The exocrine HOTAIR promotes the proliferation, migration and invasion of endothelial cells, inhibits endothelial cell apoptosis, and thus promotes the development of endometriosis. It is also an important mechanism of disease progression to regulate target cells by transferring long chain non coding RNA (lncRNA) from exosomes. Antisense hypoxia inducible factor (aHIF) is an lncRNA related to angiogenesis, and aHIF is highly expressed in ectopic endometrium and serum exosomes of patients with clinical endometriosis. Exosomes aHIF transfer from endometrial stromal cells to HUVECs, and then activate vascular endothelial growth factor (VEGF)–A, VEGF–d and basic fibroblast growth factor, It induces the angiogenesis behavior of HUVECs, thus promoting the progress of endometriosis (*Qiu et al., 2019*). miR-126 is an endothelial cell-specific expressed miRNA with important functions in the control of vascular integrity and angiogenesis (*Wang & Olson, 2009*). *Dai, Gu & Di (2012)* suggested that miR-199a-5p promoted the proliferation, migration and angiogenesis of Ec-MSCs by regulating the activity of the NF-κB pathway and the production of interleukin 8 (IL-8). In addition, it

has been found that miR-210 expression is upregulated in EcESCs. miR-210 can induce the differentiation of Eu-ESCs into Ec-ESCs through activation of STAT3, which is involved in cell proliferation, anti-apoptosis and angiogenesis (*Okamoto et al., 2015*).

## EVs enhance cell proliferation, migration and invasion

Exosomal miR-22-3p, an important regulator of cell proliferation, was significantly increased in the exosomes of abdominal macrophages from patients with endostosis, and the functional analysis of miR-22-3p by knockdown and overexpression of miRNAs showed that the upregulation of exosomal miR-22-3p induced an increase in the expression of nuclear factor-κB (NF-κB) (*Zhang et al., 2020a*). The results of this study showed that exosomal miR-22-3p in abdominal macrophages may enhance the proliferation, migration and invasion of Ec-ESCs by regulating the SIRT1/NF-κB signalling pathway.

## EVs are associated with immune response and chronic inflammation

In addition, endometriosis is an inflammatory disease whose pathogenesis has been linked to peritoneal macrophages, and exosomes have been implicated as key regulators of several inflammatory diseases. A study found elevated levels of small heat shock protein (sHsp)-22 in exosomes of patients with EMS, and the level of heat shock protein expression was positively correlated with the expression of markers of cytotoxic immune response, *i.e.,* perforin and granzyme B, suggesting a link between exosomes and immune imbalance in EMS (*Wyciszkiewicz et al., 2019*). A study by *Chen et al. (2019)* concluded that exosomal miRNAs derived from myeloid-derived suppressor cells (MDSCs) may contribute to the development of EMS by suppressing the immune function of the organism, suggesting that exosomes play an important role in the process of immune imbalance in EMS.

*Sun et al. (2019b)* injected EMS-exosomes derived from ectopic stromal cells into mice with endometriosis model, while the control group received con-exosomes from mice without endometriosis. Finally, the peritoneal macrophages were evaluated and the ectopic lesions were counted and measured. It was found that after the treatment of EMS-exosome, the macrophages polarized into m2-like phenotype, and their phagocytosis decreased. Mice in the EMS-exosome group showed a significant increase in total lesion weight and volume and enhanced m2-like macrophage infiltration in the EMS-exosome group compared with the con-exosome group. This emphasizes the importance of EMS-exosomes in the pathogenesis of endometriosis. *Huang et al. (2022)* found that the endo-heteroplasia-derived exosome miR-301a-3p may induce M2-type macrophage polarisation by up-regulating PI3K expression and down-regulating PTEN expression through the phosphatase and tensin homolog (PTEN)-PI3K axis. The above studies suggest that exosomes transferred from ESCs to macrophages contribute to immune dysfunction by inducing the conversion of macrophages to an M2-like phenotype and attenuating their phagocytic capacity. Also, exosomes derived from ESCs promote infiltration of M2-like macrophages and increase the size and weight of ectopic lesions in mice. Figure 2 summarizes the information transfer among ESCs, macrophages, and MSCs through EVs. The above experiments, although they may not fully mimic human physiology, may, to a certain extent, also indicate that extracellular

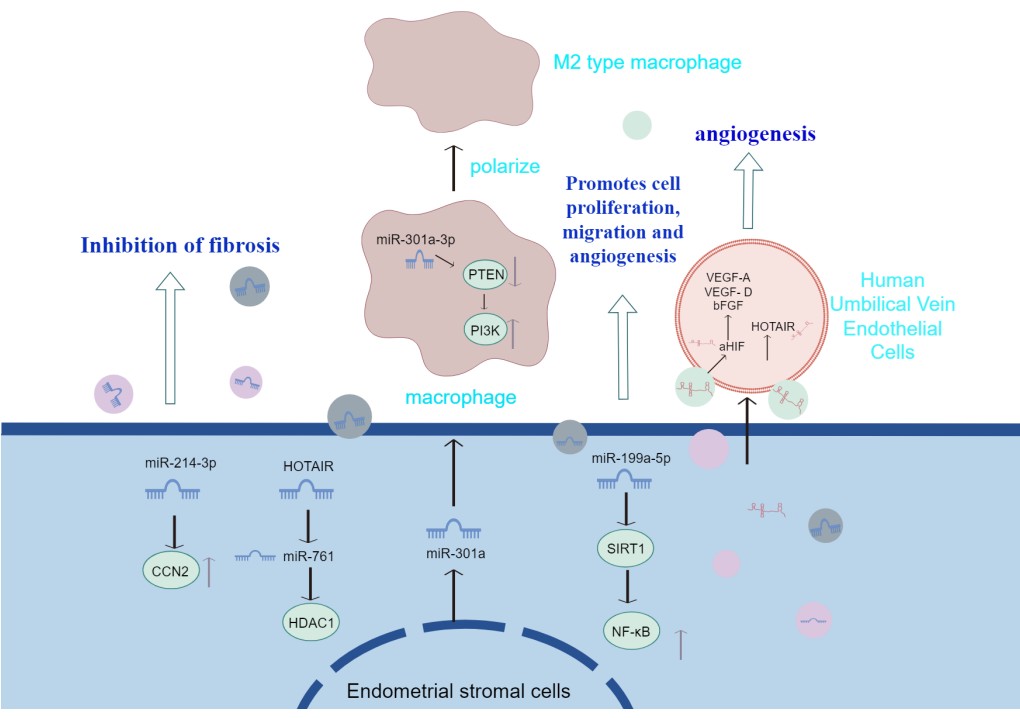

**Figure 2** **The information transfer among ESCs, macrophages, and MSCs through EVs.** Effective containment of endometrial fibrosis, promotion of proliferation, migration and angiogenesis of Ec-MSCs by modulation of cytokines or activation of signalling pathways in extracellular vesicles of endometrial stromal cell origin; and transfer from ESCs to macrophages, which contributes to immune dysfunction by prompting conversion of macrophages to an M2-like phenotype and attenuating their phagocytic capacity.

vesicles identifying macrophage targets and altering macrophage polarisation type and function in therapy will have a positive impact on the treatment of endometriosis.

## Infertility
### EVs are involved in the regulation of inflammation
Extracellular vesicles can be released from the endometrial epithelium into the uterine cavity, and these vesicles contain specific miRNAs, which can be transferred to blastocyst trophoblast ectodermal cells or endometrial epithelial cells to promote implantation, for example, trophoblast-derived exosomes transferred to endometrial epithelial cells can promote epithelial-mesenchymal transition and endometrial cell migration, and will be involved in the regulation of inflammation. Screening for specific miRNAs in these exosomes resulted in specific enrichment of MiR-1290 in exosomes. miR-1290 promoted the expression of inflammatory factors (interleukin IL-6 and IL-8) and endometrial epithelial cell migration, and exosomal miR-1290 promoted angiogenesis *in vitro* (*Shi et al., 2021*). *Taylor et al. (2020)* also found that the isolation of EVs from trophoblast cells enhanced the release of inflammatory cytokines from metaphase cells, which is considered to be an important component of a successful pregnancy.

### Extracellular vesicles influence embryo development and implantation

The transcriptome of uterine fluid-derived extracellular vesicles, on the other hand, is closely related to genes regulating cell adhesion and attachment and to the endometrial tissue transcriptome (*Giacomini et al., 2021*). Heteronuclear ribonucleoprotein C1 (hnRNPC1) is involved in the internalization of endometrium miR-30d into the exocrine body, preparing for its subsequent integration into trophoblastic ectodermal cells (*Balaguer et al., 2018*). It was found that uptake of human endometrial miR-30d by mouse blastocysts resulted in overexpression of Itb3, Itga7 and Cdh5 genes, which are associated with embryo adhesion, and that they increased the rate of embryo adhesion. Greening discovered that human trophoblast cells ingest exosomes, which improves their adherence. This reaction is partially mediated by active focal adhesion kinase (FAK) signaling (*Greening et al., 2016*). Metalloproteinases such as matrix metalloproteinases (MMP)-14 and A Disintegrin and Metalloproteinase 10 (ADAM 10) are present in endometrially derived exosomes. Exosomes MMP-14 and ADAM10 can play key roles in the process of implantation (*Latifi et al., 2018*). The exosomes ADAM10 and MMP-14 can influence the bioavailability of various factors involved in embryo implantation at the embryo-maternal interface. One of the most likely roles of exosome MMP-14 is to degrade the extracellular matrix in the endometrial space of the embryo, thus preparing the right conditions for the embryo to approach the endometrium (*Alameddine, 2012*). Exosome ADAM 10 could be involved in the release of soluble HB-EGF from endometrial cells, which could then attach to the appropriate receptor on the embryo (probably ErbB 4) and influence embryo development and implantation (*Chobotova et al., 2002*). In addition, extracellular vesicular miR-21-5p derived from follicular fluid regulates apoptotic proteins during cellular communication by targeting the PI3K/AKY and JAK/STAT3 signalling pathways, which play a dynamic role in pre-implantation embryo development (*Khan et al., 2021*). A new study on bovine embryos has found that EVs secreted during blastocyst development regulate gene expression in bovine endometrial cells (*Aguilera et al., 2023*).

### EVs increase endometrial tolerance

In addition, trophoblast-derived exosomes (TDEs) can have an effect on the endometrial acceptability of endometrial epithelial cells (EECs) endometrium, *Su et al. (2022)* investigated the mechanisms by which trophoblast-derived exosomes and their proteins are involved in fetal-maternal interactions during bovine pregnancy. They co-cultured endometrial epithelial cells with trophoblast-derived exosomes from progesterone-treated bovine trophoblast cells and found that the treatment enhanced the expression of endometrial acceptability factors, integrins αv, β3, Wnt7a, and MUC1, compared with the untreated endometrial epithelial cells, by altering their extracellular environment, metabolism, and redox homeostasis through proteomic comparisons. One of the most important components of fetal-maternal interactions is trophoblast-derived exosomal proteins; modifications to these proteins could be a crucial signal for modifying endometrial receptivity and offer a possible target for enhancing fertility (*Su et al., 2022*). Exosomes loaded with human chorionic gonadotropin (hCG) had a greater effect on endometrial tolerance than hCG or exosomes alone. hCG upregulated LIF and trophoblast genes,

and downregulated Mucin-16 (Muc-16) and insulin-like growth factor binding protein 1 (IGFBP1) genes. When used as hCG-loaded exosomes, the effects of hCG on leukemia inhibitory factor (LIF) and Muc-16 expression were significantly enhanced (*Hajipour et al., 2021*). Therefore, in the human uterine microenvironment, exosomes support endometrial-embryonic interactions, which are essential for a successful implantation. It has been demonstrated that human umbilical cord mesenchymal stem cell (HUMSC)-exosome gels significantly increase thin endometrial tolerance and improve pregnancy rates by accelerating subendometrial microangiogenesis *via* the TGF-β1/smad2/3 signaling pathway and preventing the suppression of endometrial fibrosis (*Zhang et al., 2022b*).

In addition, ecdysis has a crucial role in the embryo implantation process. Ecdysis is the transformation of endometrial mesenchymal fibroblasts into specialised secretory ecdysis cells, which provide nutrients and immunomodulatory substrates for embryo implantation and placenta development. It was found that miR-294 in embryonic stem cell-derived exosomes promotes cell proliferation and inhibits differentiation, suggesting that ESCs-derived EXOs may promote the process of endometrial metaphase (*Wang et al., 2013*; *Guo et al., 2015*). *Cheung et al. (2021)* found that endometrial stromal fibroblasts formed by the stimulated transformation of Induced pluripotent stem cells (iPSCs) responded to signals associated with metaphase, and further found that iPSCs supernatant supernatant of iPSCs, could also promote the expression of ecdysis signals.

## Intrauterine adhesion

Intrauterine adhesions, also known as Asherman's syndrome, are primarily fibrous tissue that forms after damage to the endometrium and can arise as a result of gynecologic surgery or as a complication of pregnancy. With the rise of cesarean sections and endometrial surgery, Asherman's syndrome has become a growing problem. Due to partial or complete occlusion of the patient's uterine cavity, this in turn can lead to a variety of complications such as decreased menstruation, amenorrhoea, incomplete pregnancy or recurrent miscarriages (*March, 2011*). Modern research has found that exosomes can be used to treat intrauterine adhesions.

### *EVs from different MSC sources in the treatment of uterine adhesions*

Mesenchymal stem cells (MSCs) have non-immunogenic, angiogenic, anti-fibrotic, anti-apoptotic and anti-inflammatory properties and support tissue repair by secreting various factors and chemokines in cell therapy. *Saribas et al. (2020)* found that MSC and exosome treatment enhanced MMP-2 and MMP-9 expression. Tissue inhibitor of metalloproteinase 2 (TIMP-2) expression was decreased, and proliferation and vascularization of the uterus increased and fibrosis decreased due to MSC and exosome treatment.

On the one hand, MSCs have direct access to exosomes, reducing the production of pro-inflammatory factors and increasing the expression of anti-inflammatory factors. This process involves macrophage polarization. *Xin et al. (2020)* constructed EVs and collagen scaffolds (CS/Exos) that promoted CD163+ M2 macrophage polarisation, reduced inflammation and increased anti-inflammatory responses *in vivo* and *in vitro*, demonstrating that CS/Exos treatment promotes endometrial regeneration and fertility

restoration, possibly through the immune-modulatory function of miRNAs. Others have found that tumor necrosis factor-α (TNF-α)-triggered mesenchymal stem cells (T-MSCs) EVs promote endometrial M1-type macrophage polarisation to the M2 phenotype *via* the Jak-STAT signalling pathway, mainly *via* galactoglucan lectin-1 in the T-MSCs-EVs, in order to exert anti-inflammatory effects of EVs in IUA (*Li et al., 2022*). The above studies illustrate that EVs play an important role in the inflammatory response. On the other hand, exosomes isolated from MSCs can significantly reverse endometrial fibrosis through a series of signaling pathways including transforming growth factor-β (TGF-β), Wnt, Snail and Hippo. *Xiao et al. (2019)* found that miR-340 transferred from bone marrow mesenchymal stem cells inhibits TGF- β 1. Up regulation of fibrosis gene expression in ESCs. These data indicate that the effective anti fibrosis function of bone marrow mesenchymal stem cells can transfer miR-340 to endometrial stromal cells through exosomes, while enhancing the transfer of miR340 derived from bone marrow mesenchymal stem cells is another way to prevent intrauterine adhesions. Another study found that menstrual blood-derived stromal cell-derived EVs (MenSCs-sEVs) restored the normal morphology of the uterine cavity, allowing angiogenesis and glandular regeneration, and allowing the endometrial fibrosis to be reversed (*Zhang et al., 2021a*). EVs derived from placental mesenchymal stem cells (PMSC) were found to regulate the TGF-β/smad pathway *via* miR-125b-5p, miR-30c-5p and miR-23a-3p to exert a role in promoting cell proliferation, increasing the thickness of the endometrium, and reversing fibrosis, which could repair the damaged endometrium of the animals and enhance their fertility (*Liu et al., 2024*). It has also been found that exosomes derived from the novel mesenchymal cell Telocytes (TC-Exo) play an important role in endometrial regeneration and the reversal of endometrial fibrosis. TC-Exo treatment resulted in significant endometrial regeneration, enhanced angiogenesis, reduced fibrosis, increased number of glands, significant increase in endometrial thickness, MVD and VEGF, reduced fibrotic area, and reduced FN expression within the endometrium (*Chen et al., 2023*). Not only that, Menstrual blood-derived stromal cell-derived exosomes (MenSCs-EXO) also restore IUA endometrial morphology, reduce fibrosis, and promote endometrial and vascular proliferation *via* UBR4-mediated YAP activity (*Qi et al., 2023*).

In addition, modification of MSCs can produce specific exosomes and enhance angiogenesis (*Wu et al., 2018b*; *Zhang et al., 2020c*; *Bian et al., 2022*). The core of the therapeutic efficacy of MSCs lies in their potential to release paracrine factors, a process mediated through MSCs-derived exosomes, particularly adipose tissue-derived MSC exosomes, *i.e.,* adipose mesenchymal stem cell-derived exosomes (ADSC-exo), *Zhao et al. (2020)* delved into the therapeutic potential of ADSC-exo in a rat model of intrauterine adhesions (IUA). It was found that in the IUA model, the application of ADSC-exo was effective in preserving the structural integrity of the uterus, triggering the construction of the neovascular network and tissue repair mechanisms, as well as curbing the process of local fibrosis. In addition, the treatment significantly promoted the regeneration process and collagen fibre rearrangement in the endometrium, and up-regulated the expression levels of key molecules such as integrin-β3, leukaemia inhibitory factor (LIF) and vascular endothelial growth factor (VEGF). This series of effects not only optimised the receptive environment of the regenerating endometrium, but also highlighted the unique ability

of ADSC-exo in promoting endometrial remodelling, enhancing endometrial tolerance and restoring reproductive function. Thus, ADSC-exo brings a new therapeutic light as a potential treatment for the patient population suffering from IUA and infertility, and is expected to significantly improve their fertility prospects (*Zhao et al., 2020*). *Zhu et al. (2022)* found that bone marrow mesenchymal stem cells (BMSCs)-derived EVs-treated rats promoted endometrial angiogenesis and regeneration by up-regulating the JAK/PI3K/mTOR/STAT3 signalling pathway, suggesting that Cardiotrophin-1 (CTF1)-modified BMSCs-exo promotes endometrial angiogenesis and regeneration, thereby facilitating uterine repair after injury. However, current research on the relationship between extracellular vesicles and uterine adhesions relies heavily on animal experimental models that may not fully mimic human physiology. How to verify that this finding can be applied to humans will be a direction for further research in the future.

In addition, it has been noted that combining ADSC-Exo with some biological materials can more significantly promote endometrial regeneration, which is more conducive to tissue regeneration and repair and disease treatment (*Lin et al., 2021*). Human bone marrow mesenchymal stem cell-derived exosomes, or BMSC-Exo, also aid in the healing of endometrial damage sustained after IUA therapy. CD9, CD63, and CD81 are exosome-specific proteins that were expressed by BMSCs-Exo. The contents of BMSCs Exo can enter target cells. Exo BMSCs have the ability to heal injured endometrium in an IUA model and stimulate cell migration and proliferation *in vitro*. The presence of miR-29a in BMSCs-Exo may play a significant role in its resistance to fibrosis during endometrial healing in IUA (*Tan, Xia & Ying, 2020*).

## Endometrial cancer

MIR210HG acts as an oncogene to promote proliferation and metastasis in endometrial cancer cells and regulates malignant development of endometrial cancer cells partly through spongy miR-337-3p and miR-137, thus regulating the level of high mobility group protein A 2 (HMGA2) (*Fan et al., 2021*). Another study found that circRNAs in extracellular vesicles isolated from sera of patients with endometrial cancer play an important role, and that the main pathway for these circRNAs is the isolation of cancer-mediated miRNAs. circ 0109046 and circ 0002577 reached a greater than 2-fold fold fold change when using real-time quantitative PCR (*Xu et al., 2018*).

Endometrial cancer is the second most common cancer in the female reproductive system and the sixth most common cancer in women (*Sung et al., 2021*). Abnormal uterine bleeding is the most common symptom of endometrial cancer (*Amant et al., 2005*). Cancer cells interact closely with neighboring stromal cells and jointly promote disease through bidirectional communication. A large number of studies have found that exosomes play a communication role between cancer cells and neighboring fibroblasts. Exosome-derived miRNAs regulate the adhesion, invasion, and angiogenic capacity of endometrial cells and modulate endometrial cell proliferation and apoptosis. *Yan et al. (2014)* found for the first time that miR-302 cluster could directly target Cyclin D1 and indirectly regulate the expression of cell cycle protein-dependent kinase 1 gene, thereby inhibiting the proliferation and migration of endometrial cell lines Ishikawa and HEC-1B, and blocking the cell cycle

at G2/M phase, inhibiting proliferation and differentiation as well as decreasing the tumourigenicity of cancer cells. A large number of studies have found that exosomes play a role in communication between cancer cells and neighbouring fibroblasts. Exosome miR-320a secreted by cancer-associated fibroblasts is directly transferred to endometrial cancer cells and can inhibit cancer cell proliferation (*Zhang et al., 2020b*).

*Maida et al. (2016)* isolated endometrial fibroblasts from normal endometrial tissues and endometrial cancer tissues and examined the intercellular transfer of exosomes originating from the endometrial cancer cell line Ishikawa, and showed that exosomes secreted by endometrial cancer cells were accepted by endometrial fibroblasts, and these exosomes were able to transport functional small RNAs to fibroblasts, demonstrating for the first time that endometrial cancer cells communicate with neighbouring fibroblasts by transferring small regulatory RNAs *via* exosomes, and suggesting that tumor cells may regulate the tumor microenvironment through the contents of exosomes which are mainly miRNAs. *Srivastava et al. (2018)* have searched for hsa-miR-200c-3p from exosomes of urinary origin for the diagnosis of endometrial cancer. Therefore using liquid biopsy technique, uterine lavage fluid to search for exosomes as an early marker for endometrial cancer is more potential development. *Roman-Canal et al. (2019)* successfully extracted EVs from the peritoneal lavage fluid of endometrial cancer patients and healthy individuals, and then analysed the miRNA content of these EVs in detail. The results of the study showed that there was a significant imbalance in the expression levels of 114 miRNAs in the diseased group compared to the control group. Accordingly, they hypothesised that such EV-associated miRNA signature profiles may be a key source of biomarkers for early cancer detection. *Li et al. (2019)* found that miR-148b could be transferred from cancer-associated fibroblasts (CAFs) to endometrial cancer cells *via* exosomes and inhibit endometrial cancer metastasis by directly binding to its downstream target gene, *DNA methyltransferase 1 (DNMT1)*, thus acting as an oncogene. In endometrial cancer, *DNMT1* plays a potential role in enhancing cancer cell metastasis by inducing epithelial-mesenchymal transition (EMT). Therefore, down-regulation of miR-148b induces EMT in endometrial cancer cells, thereby alleviating the inhibitory effect of *DNMT1*. However, it has also been found that exosomal nuclear enriched abundant transcript 1 (NEAT1) from cancer-associated fibroblasts promotes endometrial cancer progression through the miR-26a/b-5p-mediated STAT3/YKL-40 pathway (*Fan et al., 2021*). *Shi et al. (2020)* purified exosomes from endometrial cancer cell culture medium and found that endometrial cancer cells could secrete exosomes containing miR-133a, confirming that miR-133a may regulate the down-regulation of the FOXL2 (forkhead box L2) gene in endometrial cancer tissues thereby inhibiting tumor cell proliferation and increasing apoptosis.

The exosomes in plasma also play a key role in the pathophysiology of endometrial carcinoma. Plasma derived exocrine miR-15a-5p is a promising and effective diagnostic biomarker for early detection of endometrial carcinoma (*Zhou et al., 2021*). Another study discovered that as EC progressed, the plasma exosomal lectin galactoside-binding soluble 3 binding protein (LGALS3BP) rose indicating that human umbilical vein endothelial cells (HUVEC) function through the activation of the PI3K/AKT/VEGFA signaling pathway both *in vitro* and *in vivo*, and that highly contained exosomal LGALS3BP contributed to

EC cell migration and proliferation. It also suggested that plasma exosomes containing LGALS3BP contributed to EC growth and angiogenesis during EC progression, offering a new insight into EC diagnosis and prognosis (*Song et al., 2021*).

There are also several studies that have identified tumor-associated macrophages (TAM) as a major regulator of endometrial cancer (EC) development, and TAM can crosstalk with cancer cells by transferring exosomes carrying microRNAs (miRNAs) (*Wang et al., 2022*). Exosomes from CD45ROCD8 T cells have been isolated to examine their regulatory role in endometrial cancer. miR-765 was found to be the most severely down-regulated miRNA in the UCEC genes, and these exosomes limit endometrial cancer disease progression by regulating the miR-765/PLP2 axis (*Fan et al., 2021*).

Therapeutically, exosomes can be used as drug carriers to participate in tumor therapy, and targeting exo-miRNAs to treat tumors is a novel and effective therapeutic modality. Studies have shown that a single miRNA can target hundreds of mRNAs at the same time, affecting cellular biological functions by regulating the post-transcriptional activity of multiple target mRNAs, suggesting the feasibility of targeting miRNAs as a therapeutic tool (*Shah et al., 2016*). *Song et al. (2022)* analysed SERPINA5 expression in EC patients using The Cancer Genome Atlas (TCGA) database. Compared with EC patients without distant metastases and the normal population, SERPINA5 expression was significantly lower in EC patients with distant metastases ($P < 0.01$), while overexpression of SERPINA5 or high exosomal SERPINA5 levels inhibited the metastatic potential of EC by suppressing the activation of integrin β1/FAK signalling pathway. It is suggested that exosomes play an important role in tumor immunity.

## Endometritis

Chronic endometritis is a persistent inflammatory disease characterized by plasma cell infiltration in the endometrial stromal area (*Kimura et al., 2019*), with clinical manifestations including amenorrhoea, scanty menstruation, infertility, or recurrent miscarriages (*Kelleher et al., 2018*), which seriously affects women's health and fertility. MiR-27a-3p and miR-124-3p have been found to be up-regulated in endometrium and serum affected by chronic endometritis, suggesting that miR-27a-3p and miR-124-3p may represent non-invasive markers of CE and could be used in the future to assess endometrial quality in IVF (*Di Pietro et al., 2018*).

## The role of endometrial epithelium-derived exosomes in endometritis

In one study, 118 differentially expressed miRNAs were found in the uterine fluid secretions of healthy cows and patients with endometritis, of which 52 miRNAs were down regulated and 66 were up regulated. In addition, compared with IVF embryos co-incubated with healthy exosomes, IVF embryos co-incubated with endometritis exosomes significantly reduced the blastocyst formation rate. Therefore, the exosomes miRNA may be the cause of infertility due to endometritis (*Wang et al., 2019*). miR-218 in endometrial epithelial cell (EEC)-derived exosomes has also been implicated in the pathogenesis of bovine endometritis. Reduced levels of miR-218 in EEC-derived exosomes, when transferred to placental trophoblast cells, affects embryonic development and inhibits migration

of placental trophoblast cells by targeting secreted curl related protein 2 (*Wang et al., 2021*). Exosomes released by endometrial epithelial cells, which contain miR-218, exhibit a powerful anti-inflammatory potential by effectively blocking the activity of a series of immunomodulatory factors and chemokines. These exosomes act as natural delivery vehicles that can precisely deliver miR-218 into the uterine microenvironment and its neighbouring target cells, thereby finely regulating the local immune response. Specifically, miR-218 maintains the internal immune homeostasis of the uterus by inhibiting the expression of various key immune molecules, including interleukin-6 (IL-6), interleukin-1β (IL-1β), tumor necrosis factor-α (TNF-α), and chemokines, such as macrophage inflammatory protein-1α (MIP-1α) and MIP-1β, and plays its central role as an central role of immunosuppressive factors (*Wang et al., 2020*). This process not only reveals the important role of exosomes in intercellular communication, but also provides new molecular targets and therapeutic ideas for the regulation of endometrial inflammation. Another study demonstrated that overexpression of miR-193a-3p in bovine endometrial epithelial cells (BENDs) significantly increased the expression levels of LPS-induced pro-inflammatory cytokines such as IL-1β, IL-6 and TNF-α. Therefore, it is hypothesised that inhibition of miR-193a-3p may be a potential molecular target for the future treatment of endometritis (*Yin et al., 2021*). However, the role of miR-193a-3p in endometritis has not been fully explored, and the clinical validation of its therapeutic effect on endometritis is currently very much open, and the extension of the results from specific cell lines or experimental conditions to clinical practice should be done with caution, and the mechanism of EVs on endometritis will need to be validated in further basic and clinical studies in the future.

## EVs from different MSC sources in the treatment of endometritis

MSC-derived exosomes are an important paracrine product that has been used in the treatment of inflammatory diseases due to their immunomodulatory functions and tissue repair capabilities similar to those of MSCs. An injectable hydrogel scaffold with interleukin-1β (IL-1β)-activated MSC-derived exosomes has been demonstrated for the treatment of endometritis (*Zhao et al., 2023*). In addition to this, exosomes from umbilical cord MSCs (hUCMSCs-Exo) play key roles in tissue repair, including anti-inflammatory, antioxidant, implantation-supporting, and trophic functions (*Samsonraj et al., 2017*). A previous study reported that treatment of cows with bovine adipose tissue-derived stem cells (ADSCs) suppressed levels of tumor necrosis factor-α (TNF-α), interleukin-1β (IL-1β) and interleukin-6 (IL-6) in endometrial epithelial cells, thereby inhibiting lipopolysaccharide (LPS)-induced inflammation, demonstrating the role of ADSC-Exos in the endometrial cells *via* the miR-21/TLR4/NF-κB signalling pathway. Endometrial cells overexpress miR-21 through the miR-21/TLR4/NF-κB signalling pathway, thereby decreasing the expression of TLR4, reducing the phosphorylation of p65, a downstream effector of TLR4, and exerting anti-inflammatory effects (*Wang, Li & Yu, 2023*). It suggests that ADSC-Exos has a positive effect on the treatment of chronic endometritis. ADSCs-derived Exos inhibited LPS-induced endometrial cell inflammation and mouse models (*Lu et al.,*

*2021*). Table 1 summarises the source and mechanism of action of extracellular vesicles in endometrial-related diseases.

## Technical challenges

Extracellular vesicles hold great promise in unfolding as novel biomarkers of disease, therapeutics, and drug delivery vectors; However some reports describe conflicting results, even from extracellular vesicles of the same cell type. For example, MSC-derived exosomes have been shown to both inhibit and promote tumourgrowth (*Bruno et al., 2013*; *Zhu et al., 2012*). Secondly, there are still many technical challenges regarding how to effectively utilise these small molecule messengers for future applications. Despite the clinical potential of EV, the lack of sensitive EV preparation and analysis techniques poses a barrier to clinical translation (*Konoshenko et al., 2018*). There is an urgent need for efficient and large-scale clinical-grade preparation processes, effective isolation techniques, suitable storage conditions, precise modification strategies, rigorous purification steps, and precise target delivery methods (*Sil et al., 2020*). Multiple approaches have been used to assess the biodistribution of EVs *in vivo*, including fluorescent labelling of lipids and proteins, immunofluorescence, bioluminescence, Pet, sPeCt, Mri and Ct imaging59 (*Van Niel et al., 2022*). All these methods have limitations in tracking the distribution of EVs, so new methods with large dynamic range of temporal and spatial resolution will be needed in the future to overcome these limitations.In addition, methodological issues and the heterogeneity of EV components have hindered the progress of EV validation trials and the development of EV-based diagnostic and therapeuticproducts (*González et al., 2024*). Because of the inherent complexity, size heterogeneity, and natural variation of exosomes throughout the assembly process, the application of exosomes as reliable therapeutic vectors necessitates a scalable manufacturing process that produces exosomes in a rapid, cost-effective, and reproducible manner (*Rezaie, Feghhi & Etemadi, 2022*).

## CONCLUSION

In this review, we summarize the results of extracellular vesicle research. The complex pathophysiologic functions of extracellular vesicles are closely related to the pathogenesis or prognosis of endometrial diseases. Extracellular vesicles act on the endometrium through multiple signalling pathways, and the intricacies between the various mechanistic pathways suggests a variety of possibilities for the treatment of endometrial-related diseases with EVs, and there may still be some undiscovered pathways, which provide a direction for more in-depth studies in the future. Although the mechanism of action of extracellular vesicles is still not fully defined, there is increasing evidence that extracellular vesicles can affect diseases such as endometriosis, infertility, endometrial cancer, intrauterine adhesions, and endometritis by means of fibrosis, angiogenesis, cell proliferation, and inflammatory responses. Due to the complexity of the biological processes involved in extracellular vesicles, the correlation between extracellular vesicles and the endometrium still requires further research. Because extracellular vesicles have low immunogenicity, they can deliver drugs directly to the cells therefore extracellular vesicles alter the physiological function of target cells by loading drugs have a promising future in the study of disease treatment.

**Table 1  Sources and mechanisms of action of extracellular vesicles in endometrial-related diseases.**

| Disease | Source | Studied cargo | Type | Function | Reference |
|---|---|---|---|---|---|
| Endometriosis | ESCs<br>ESCs<br>ESCs<br>EC | miR-761<br>miR-210<br>miR-199a-5P<br>miR-126 | miRNA<br>miRNA<br>miRNA<br>miRNA | Promote angiogenesis | *Sun et al. (2019a)*, *Qiu et al. (2019)*, *Wang & Olson (2009)*, *Dai, Gu & Di (2012)* |
| | Stroma-ectopic cells | miR-214-3p | miRNA | Inhibiting endometrial fibrosis | *Zhang et al. (2021b)* |
| | Peritoneal macrophages | miR-22-3p | miRNA | Enhances cell proliferation, migration and invasion | *Zhang et al. (2020a)* |
| | Ectopic endometria, ESCs | aHIF | LncRNA | Activation of vascular endothelial growth factor (VEGF)-A, VEGF-d and basic fibroblast growth factor induced angiogenic behaviour in HU-VECs | *Qiu et al. (2019)* |
| | ESCs | miR-301a-3p | miRNA | Induction of M2-type macrophage polarisation | *Huang et al. (2022)* |
| | MDSCs | miR-1908, miR-130b, miR-451a, miR-486-5p, *etc* | miRNA | Suppression of the body's immune function to promote the development of EMS | *Chen et al. (2019)* |
| Infertility | Trophoblast Cell | miR-1290 | miRNA | Promotion of inflammatory factor expression and endometrial epithelial cell migration | *Shi et al. (2021)* |
| | Follicular fluid | miR-21-5p | miRNA | Targeting PI3K/AKY and JAK/STAT3 signalling pathways to regulate apoptotic proteins with dynamic roles in pre-implantation embryo development | *Khan et al. (2021)* |
| | Endometrial fluid | miR-30d | miRNA | Increased embryo adhesion through overexpression of Itb3, Itga7 and Cdh5 genes | *Balaguer et al. (2018)* |
| | ESCs | miR-294 | miRNA | Promotes cell proliferation, inhibits differentiation and promotes the process of endometrial metaplasia | *Wang et al. (2013)*, *Guo et al. (2015)* |
| | Extravillous trophoblast cells | MMP-14, ADAM10 | Protein | Metalloproteinases in exosomes affect the bioavailability of factors involved in embryo implantation at the embryo-maternal interface | *Latifi et al. (2018)*, *Alameddine, (2012)*, *Chobotova et al. (2002)* |
| Intrauterine adhesion | PMSC | miR-125b-5p, miR-30c-5p, miR-23a-3p | miRNA | EVs derived from placental MSCs to promote cell proliferation, increase endometrial thickness and reverse fibrosis may repair the endometrium and enhance its fertility | *Liu et al. (2024)* |
| | BMSCs | miR-29a | miRNA | Inhibiting endometrial fibrosis | *Tan, Xia & Ying (2020)* |
| | MSCs | MMP-2, MMP-9, TIMP-2 | Protein | Induction of proliferation and angiogenesis, inhibition of fibrosis | *Saribas et al. (2020)* |

*(continued on next page)*

 

**Table 1** (*continued*)

| Disease | Source | Studied cargo | Type | Function | Reference |
|---|---|---|---|---|---|
| | MenSCs | — | — | Restoration of IUA endometrial morphology, reduction of fibrosis and promotion of endometrial and vascular proliferation through UBR4-mediated YAP activity | *Qi et al. (2023)* |
| Endometrial cancer | Serum | miR-320a | miRNA | Inhibition of cancer cell proliferation | *Zhang et al. (2020b)* |
| | Supernatants of endometrial cancer cell lines | miR-133a | miRNA | Regulation of FOXL2 gene down-regulation in endometrial cancer tissues to inhibit tumor cell proliferation and increase apoptosis | *Shi et al. (2020)* |
| | Urine | hsa-miR-200c-3p | miRNA | For the diagnosis of endometrial cancer | *Srivastava et al. (2018)* |
| | CAFs | miR-148b | miRNA | Inhibits endometrial cancer metastasis by binding to its downstream target gene DNMT1, thereby acting as a tumor suppressor | *Li et al. (2019)* |
| | Plasma | LGALS3BP | Protein | Activation of PI3K / AKT / VEGFA signalling pathway in vitro and in vivo promotes proliferation and migration of endometrial cancer cells | *Song et al. (2021)* |
| | CD45ROCD8 T cells | miR-765 | miRNA | Limiting endometrial cancer disease progression by regulating the miR-765 / PLP2 axis | *Fan et al. (2021)* |
| | Ishikawa and HEC-1-A | miR-133a | miRNA | Regulates down-regulation of the FOXL2 gene in endometrial cancer tissues thereby inhibiting tumor cell proliferation and increasing apoptosis. | *Shi et al. (2020)* |
| | Serum | Hsa circ 0109046, hsa circ 0002577 | circRNA | Isolation of cancer-mediated miRNAs | *Xu et al. (2018)* |
| Endometritis | plasma | miR-27a-3p miR-124-3p | miRNA | Up-regulated in endometrium and serum affected by chronic endometritis | *Di Pietro et al. (2018)* |
| | EEC | miR-218 | miRNA | Blocking immune factors and chemokines to suppress inflammation | *Wang et al. (2021)* |
| | ADSCs | miR-21 | miRNA | Overexpression of miR-21 through the miR-21/TLR4/NF-κB signalling pathway, which reduces TLR4 expression, decreases phosphorylation of the TLR4 downstream effector p65 and exerts anti-inflammatory effects | *Lu et al. (2021)* |
| | BENDs | miR-193a-3p | miRNA | Increased expression levels of LPS-induced pro-inflammatory cytokines such as IL-1β, IL-6 and TNF-α | *Yin et al. (2021)* |

**Notes.**

Abbreviations: ESCs, endometrial stromal cells; EC, endothelial cell; MDSCs, myeloid-derived suppressor cells; PMSC, placental mesenchymal stem cells; BMSCs, Bone marrow mesenchymal stem cells; MSCs, Mesenchymal stem cells; MenSCs, menstrual blood-derived stromal cells; EEC, endometrial epithelial cell; ADSCs, adipose tissue-derived stem cells; CAFs, cancer-associated fibroblasts; BENDs, bovine endometrial epithelial cells.

However, most of the current studies on extracellular vesicles and endometrial-related diseases lack systematic and multidimensionality, and the application of exosomal miRNAs in the treatment of endometrial-related diseases is still in its infancy. For example, there are still large gaps yet to be filled in the clinical validation of EV-based drug delivery. This also provides new challenges for future research.

### Funding
This work was supported by the State Administration of Traditional Chinese Medicine (National TCM Human Education Letter (2022) No. 1). The funders had no role in study design, data collection and analysis, decision to publish, or preparation of the manuscript.

### Grant Disclosures
The following grant information was disclosed by the authors:
The State Administration of Traditional Chinese Medicine (National TCM Human Education Letter (2022) No. 1).

### Competing Interests
The authors declare there are no competing interests.

### Author Contributions
- Zilu Wang conceived and designed the experiments, performed the experiments, prepared figures and/or tables, and approved the final draft.
- Feng Li performed the experiments, authored or reviewed drafts of the article, and approved the final draft.
- Wenqiong Liu analyzed the data, authored or reviewed drafts of the article, and approved the final draft.

### Data Availability
This is a literature review.

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
