# Peer review of "Extracellular vesicles in endometrial-related diseases: role, potential and challenges"

_PeerJ, doi:10.7717/peerj.19041_

## Round 0.1 · original submission · Minor Revisions

As you will, both reviewers comment favourably on the review, and recognise it has merit. Both have expressed some concerns and offer suggestions for improvement which you should take on-board. I note that one has expressed a desire to see you reflect on how you generalise to clinical practice, and one asks that you include a section on challenges faced in this fast-moving area.

·

Basic reporting

The manuscript addresses the critical role of extracellular vesicles (EVs) in endometrial-related diseases, which is a growing field of research with significant implications for diagnostics and therapeutics.The abstract should explicitly mention the methodologies employed for the review. A sentence summarizing the survey methodology could enhance clarity.Simplify the language to make it more accessible to a broader audience, especially for non-specialists.

Experimental design

Provide more details about the survey methodology in the main text. For example, the inclusion and exclusion criteria, number of studies reviewed, and timeline of the studies analyzed.

Validity of the findings

The manuscript frequently references in vitro and in vivo studies (e.g., rodent models for endometriosis and intrauterine adhesions). This suggests experimental evidence underpins many findings. However:

Some conclusions rely heavily on experimental models that may not fully mimic human physiology (e.g., rodent models of endometriosis).
Clinical validation of therapeutic applications, such as EV-based drug delivery, is limited and should be emphasized as a current gap.
The manuscript explores less-established roles of EVs in specific conditions like intrauterine adhesions and infertility. While this adds novelty:

Findings related to novel EV-associated biomarkers (e.g., miR-193a-3p for endometritis) require further validation in diverse populations and clinical settings.
Generalizing results from specific cell lines or experimental conditions to clinical practice should be approached cautiously.

Additional comments

The findings align with foundational studies on EV-mediated pathogenesis but should incorporate the latest breakthroughs for added relevance.

Reviewer 2 ·

Basic reporting

The work submitted for review covers a very important and current topic. I have several comments on the reviewed work, several aspects require clarification:

1. There are some ambiguities in the abstract: line 33: "...responses, thus positively and contributing to disease. " in what sense was "positively" used?; lines 33-34: "This review describes the impact of extracellular vesicles on endometrial-related diseases, ..." what are the vesicles derived from?'' lines 35-36: "...with a view to informing the development of endometrial-related diseases and their prognostic and therapeutic treatment..." maybe their potential prognostic and therapeutic use (or significance)?

2. Each quoted statement should be accompanied by a citation, e.g. lines: 69-72: "They are also closely related to endometrial diseases and have an inhibitory effect on endometrial diseases and promote the growth and repair of the endometrium, and there are also some related extracellular vesicles that play a promotional role in the development of endometrial diseases" - citations are missing.

3. All abbreviations should be expanded on first use.

4. In some places the text is chaotic, as if composed of unrelated fragments, e.g.: lines 104-108: " This important finding not only reveals that miR-214-3p in exosomes can effectively curb the pathological process of endometrial fibrosis by precisely regulating the expression of CCN2, but also further highlights the great potential of miR-214-3p as a biomarker in the diagnosis of endometriosis (EMS) and its broad application prospects as a novel therapeutic target in the treatment strategy of endometriosis."; lines 325-329: "Li et al. [81]found that cancer-associated fibroblasts (CAFs)-derived exosomes miR-148b inhibited metastasis of endometrial cancer by directly binding to its downstream target gene DNMT1, thus playing a tumor-suppressive role. binds to inhibit endometrial cancer metastasis and thus acts as a tumor suppressor,In endometrial cancer, DNMT1 plays a potential role in enhancing cancer cell metastasis by inducing epithelial-mesenchymal transition (EMT)." and others.

I suggest verifying the entire text once again.

5. Please ensure uniform designations of non-coding RNAs throughout the text, use the same abbreviation, e.g.: miRNA, lncRNA.

6. I suggest attaching a legend to the table and figures with an explanation of the abbreviations used. So that the figures are legible and understandable to the recipient.

7. The properties of mesenchymal stem cells are mentioned several times in the text in various places. Duplicate information can be removed.

Experimental design

8. Were year restrictions used during the literature search? Was any system adopted for excluding works for review, apart from the fact that only research works were taken into account? The literature could be supplemented with additional items from the last 5 years.

9. The authors mention the potential use of extracellular vesicles in diagnostic, prognostic and therapeutic approaches; perhaps it would be beneficial to divide the subsections into such approaches as well? Especially, since the information on the possibility of diagnostic and prognostic use of extracellular vesicles derived from endometrial cells and therapeutic use of extracellular vesicles derived from e.g. MSC in the therapy of endometrial diseases is currently mixed. This would allow for a good demarcation of when EV of endometrial origin is mentioned and when EV of other origin, e.g. from stem cells, is mentioned.

10. The authors devote a lot of space to the possibility of therapeutic use of EVs derived from stem cells, maybe it would be a good idea to prepare such a subsection?

Validity of the findings

11. Since the title of the paper indicates "challenges", it is worth separating and expanding on these issues, not only in the form of conclusions. It would be necessary to briefly indicate also what technical challenges lie before the use of EVs in practice, issues of isolation, identification, determination of their charge/quantity, what future directions of research are necessary in the field of research on endometrial diseases and the use of EVs.

---

## Round 0.2 · accepted · Accept

Thanks for attending to the issues raised effectively. I am happy to support acceptance now.

·

Basic reporting

Satisfied with the revision

Experimental design

Satisfied with the revision

Validity of the findings

Satisfied with the revision

Additional comments

Satisfied with the revision